# Activity of glucose-6-phosphate dehydrogenease and its correlation with inflammatory factors in diabetic retinopathy

Dan Liu[1], Chuchu Cheng[2], Lan Zhou[3,4], Qiqiao Zeng[3], Tao Zi[3], Gongyi Chen[3], Hongyan Sun[3], Cunzi Li[3], Jun Wang[5], Ming-Ming Yang[3]*

1 Department of Ophthalmology, The Second Clinical Medical College, Jinan University, Shenzhen, China, 2 Department of Endocrinology, The Second People's Hospital of Futian District, Shenzhen, China, 3 Department of Ophthalmology, The Shenzhen People's Hospital, Shenzhen, China, 4 Post-doctoral Scientific Research Station of Basic Medicine, Jinan University, Guangzhou, China, 5 Department of Endocrinology, The Shenzhen People's Hospital, Shenzhen, China

☯ These authors contributed equally to this work.

* yang.mingming@szhospital.com

**Data Availability Statement:** All relevant data are within the manuscript and its Supporting Information files.

## Abstract

### Purpose

This study aims to explore glucose-6-phosphate dehydrogenase (G6PD) activity in diabetic retinopathy (DR) and its correlation with inflammatory factors, elucidating the regulatory role of G6PD in DR pathology.

### Methods

A total of 151 T2DM patients were divided into three groups: diabetes without retinopathy (DNR, n = 59), non-proliferative retinopathy (NPDR, n = 46) and proliferative retinopathy (PDR, n = 49). Plasma G6PD activity was measured by a Randox G6PD kit and compared between these groups. Then the G6PD activity was correlated with inflammatory cytokines and metabolic parameters in these patients. A STZ-induced diabetic rat model was established, G6PD activity was validated by western blot and immunofluorescence staining in the retina of this model.

### Results

Plasma G6PD activity decreased in the order of DNR, NPDR and PDR groups (P<0.01). G6PD activity was negatively correlated with IL-6, IL-8, TNF-α, cholesterol, and LDL (r = -0.1625, -0.1808, -0.1865, -0.1747, r = -0.1807, P<0.05). Multiple regression analysis showed TNF-α, IL-6, and LDL were independent related factors for G6PD. Logistic regression analysis showed G6PD, triglyceride, cholesterol, IL-8, TNF-α, and macular edema were influencing factors for T2DM with DR. Western Blot analysis indicated a significant reduction of G6PD expression in the retina, and immunofluorescence staining showed distribution of G6PD especially in the retinal endothelium cell decreased.

**Funding:** This study was supported in part by Shenzhen Science and Technology Project (No. JCYJ20220818102603007), and the General Project of the Shenzhen Natural Science Foundation (No. JCYJ20210324113808023 and JCYJ20220530152813030) Ming-Ming Yang received each award. YMM designed the study and revised the manuscript.

**Competing interests:** The authors have declared that no competing interests exist.

## Conclusion

G6PD may play an important role in DR occurrence and progression, with decreased expression correlating closely with lipid metabolism and inflammatory factors.

## 1. Introduction

With the growing prevalence of diabetes worldwide, the incidence of diabetic eye diseases is expected to increase as well [1–3]. Diabetic retinopathy (DR) is one of the most common microvascular complications of diabetes, [4, 5] and is also a leading cause of vision impairment in working-age adults globally [2, 3, 6]. Presently, numerous hypotheses exist regarding the occurrence and progression mechanisms of DR, including polyol pathway activation, advanced glycation end products, oxidative stress, chronic inflammatory response, and cell apoptosis. Among these, the most representative mechanisms are glucose metabolism disorder and oxidative stress, with glucose-6-phosphate dehydrogenase (G6PD) being a key factor in both [7–10].

The major function of G6PD is providing reduced nicotinamide dinucleotide phosphate (NADPH), which regulates glutathione recycling and helps to eliminate reactive oxygen species (ROS) from cells [11]. Endothelial cells with decreased G6PD activity and lower NADPH levels show greater susceptibility to oxidative stress [12]. However, the role of G6PD in DR and its correlation with disease severity remains unknown. Therefore, this study aims to reveal the regulatory role of G6PD in diabetic retinopathy (DR) by analyzing G6PD expression changes in DR patients' plasma and rats' retinas. We further correlated G6PD activity with inflammatory response in DR.

## 2. Materials and methods

### 2.1. Recruitment of clinical cases

The recruitment period for this study is from 01/01/2022 to 01/31/2024. We enrolled 59 subjects with type 2 diabetes without retinopathy (DNR) and 46 subjects with type 2 diabetes with non-proliferative retinopathy (NPDR) and 49 subjects with type 2 diabetes with proliferative retinopathy (PDR) in this cross-sectional study. The presence of acute infections, coronary artery disease, acute complications of diabetes mellitus, severe hepatic and renal insufficiency, cancer, rheumatic and immune system diseases, and pregnancy were exclusion criteria in this study. Written informed consent was collected from each patient according to the Declaration of Helsinki and was approved by the ethical committee of the institute. We investigated the glycemic statuses of all subjects with diabetes by measuring fasting and postprandial blood glucose levels and performing the fructosamine test. DR was diagnosed by dilated fundus examination with slit lamp biomicroscopy by ±90D and 3-mirror lens 7-field digital fundus photography with fluorescence angiography. The grading of retinopathy was performed according to the modified Early Treatment Diabetic Retinopathy Study.

### 2.2. General data acquisition

General data for each patient was collected including age, gender, height, weight, and blood pressure. Body mass index (BMI) is a common index of obesity and is calculated from weight and height squared ($kg/m^2$). Measurement of serum cholesterol levels, triglycerides, and lipoproteins. Serum levels of total cholesterol and triglyceride were measured using the enzymatic

colorimetric method, and high-density lipoprotein (HDL) and low-density lipoprotein (LDL) levels were measured using the homogenous enzymatic colorimetric assay. All tests were performed on a fully automated clinical chemistry analyzer.

## 2.3. Estimation of G6PD activity

Erythrocyte G6PD activity was determined by measuring the rate of absorbance change at 340 nm due to NADP+ reduction. The rate of absorbance change was measured spectrophotometrically using the Randox G6PD kit. Data on laboratory information were also collected including fasting plasma glucose (FPG), hemoglobin A1C, and creatinine. Estimated glomerular filtration rate (eGFR) was calculated from creatinine using the MDRD equation.

## 2.4. Streptozotocin-induced diabetic rat

Twenty Sprague-Dawley (SD) rats (aged 8 weeks old, male) were obtained from Dean Gene Technology. After stable feeding, they were randomly divided into control (n = 10) and diabetic groups (n = 10). The diabetic rats were fed with high fat and sugar diets for 3 weeks (Cat#L202301056, Dean Gene Technology, China) and then fasted for 6 hours before streptozotocin (STZ, Cat#S0130, Sigma Aldrich, USA) injection. They received a single dose of STZ (75 mg/kg, freshly diluted in 0.1% mol/L citrate buffer, pH 4.5) by intraperitoneal injection. The same volume of citrate buffer without STZ was used for the control group. The fasting blood glucose was measured by using a glucometer (Roche, Switzerland) after 5 days after injection. Rats with glucose levels >15 mmol/L were considered successfully established. All animal had free access to food and water and were kept in an air-conditioned room with a 12-hour light-dark cycle. All procedures were approved by the Committee of Ethics on Animal Experiments of our hospital.

## 2.5. Western blot analysis

For western blotting, Protein was harvested from diabetic rat retinal tissue at 4°C with RIPA buffer (Cat#DB258-500, MIKX life, China) containing protein phosphatase inhibitor mixture (Cat#DB615, MIKX life, China), and the protein concentration was determined by the BCA protein assay (Cat#DB307, MIKX life, China). Protein samples were separated with 10% SDS–PAGE in a running buffer and transferred to PVDF membranes with a transfer buffer. After blocking in 5% skim milk, the membranes were incubated with primary antibodies against G6PD (1:1000, Cat#32301, SAB Signalway Antibody, China) or β-actin (1:5000, Cat#AC026, ABclonal, China) in TBST containing 5% bovine serum albumin, followed by incubation with secondary antibody for 1 h at room temperature. After capturing, the density of target protein was quantified with Image J software (National Institutes of Health, Bethesda, MD, USA) by a person who was blinded to the experiment. Levels of proteins of interest were normalized to β-actin.

## 2.6. Immunofluorescence staining

Rats were sacrificed 12 weeks after STZ injection, and their eyes were dissected and fixed in 4% paraformaldehyde overnight followed by dehydrated in 20% sucrose dissolved in PBS at 4°C overnight. After the cornea was removed and the lens was extracted, the eye was embedded into the optimal cutting temperature (OCT) compound and was then sectioned on a cryostat at 15 μm. For immunofluorescence analysis, the sections were permeabilized with 0.5% Triton X-100 for 30 min and blocked with 5% FBS for 2 h at room temperature. The samples were then incubated with primary antibodies including anti-G6PD antibody (1:1000,

Cat#32301, SAB Signalway Antibody, China) and anti-CD31 antibody (1:1000, Cat#550274, BD Pharmingen, USA) at 4°C overnight. The following day, the samples were treated with Alexa FluorTM Plus 555 or Alexa FluorTM Plus 488-conjugated secondary antibody and DAPI for laser scanning confocal microscopy (Leica, Germany) observation.

## 2.7. Immunofluorescence colocalization analysis

Analysis of colocalization between G6PD and CD 31 was performed according to previous study [13]. Briefly, a plugin named Colocalization Finder (http://questpharma.u-strasbg.fr/html/colocalization-finder.html) was inserted into image J (NIH, USA). The color images of G6PD and CD 31 were open and split into two channels in image J. Based on pixel-wise methods in Colocalization Finder, the intensity of a pixel in one channel is evaluated against the corresponding pixel in the second channel of a dual-color image, generally producing a scatterplot from which a correlation coefficient is determined.

The mice whose retinas were collected in the experiment were intraperitoneally injected with pentobarbital sodium (0.1ml/g) for anesthesia. Finally, mice were euthanized by intraperitoneal injection of pentobarbital sodium (200 mg/kg) at the conclusion of the experiment. Symptoms such as pain, weight loss, loss of appetite or weakness were set as humane endpoints for the present study; however, no animal was sacrificed before the completion of the experiment as a result of displaying any of these symptoms.

## 2.8. Statistical analysis

Data are presented as mean ± SD. We used one-way analysis of variance (ANOVA) to compare the means of different variables and observe whether there are significant differences in G6PD activity among the three groups: DNR, NPDR, and PDR. Spearman's nonparametric correlation was used to assess relationships between two variables. Linear regression analysis was performed on different study groups to calculate the square of the correlation coefficient ($R^2$) and goodness-of-fit of the model, which represents the proportion of variability in the dependent variable explained by its linear relationship with the independent variable(s). Through the results of linear regression analysis, we can further investigate which factors in the body are influenced by G6PD activity, as well as whether the occurrence and progression of DR are related to G6PD, in order to explore the possible pathways through which G6PD affects the development of DR. $P$<0.05 was considered statistically significant. All statistical analyses were performed using R version 3.5.3.

## 2.9.Ethics approval

The present study protocol was reviewed and approved by the Institutional Review Board of the Shenzhen People's Hospital and the ethics committees of all participating facilities (approval ID LL-KY-2021171). The procedures used and the care of animals were approved by the Laboratory Animal Ethics Review Board of SUSTech (approval SUSTech-JY202102017).

## 3. Results

### 3.1. G6PD activity decreased significantly in DR patients

In this study, there was no statistically significant difference in age, gender distribution, glycated hemoglobin, fasting blood glucose, alanine aminotransferase, and aspartate aminotransferase among the diabetic retinopathy (DNR), non-proliferative diabetic retinopathy (NPDR), and proliferative diabetic retinopathy (PDR) groups ($P$>0.05). Systolic blood pressure, triglycerides, cholesterol, and low-density lipoprotein were significantly higher in patients with

**Table 1. Comparison of clinical data and laboratory indicators in each group.**

| Group | n(MF) | Age | BMI | SBP | HBP | CSME(Y/N) | HbAlc |
|---|---|---|---|---|---|---|---|
| DNR | 56(37/19) | 53.39±14.81 | 24.82±3.16 | 122.34±22.11 | 78.52±8.59 | 0/56 | 9.08±270 |
| NPDR | 46(28/18) | 62.35±9.64[a] | 23.27±2.74[a] | 134.00±14,46[a] | 79,35±9.09 | 9/37[a] | 8.64±2.03 |
| PDR | 49(31/18) | 34.06±9.92[b] | 22.67±4.29[b] | 138.86±16.59[a] | 84.88±9.43[ab] | 22/27[ab] | 8.06±2.00 |
| P | 0.8615 | 0.0001 | 0.002 | 0.0000 | 0.0029 | 0.0226 | 0.1322 |
| Group | FPG | TG | TC | LDL | HDL | [a]ST | [a]LT |
| DNR | 8.17±3.12 | 1.32±0.64 | 4.15±1.07 | 2.44±0.85 | 1.06±032 | 20.36±11.68 | 23.46±17.48 |
| NPDR | 7.99±3.01 | 2.29±1.76[a] | 5.11±1.39[a] | 3.12±1.22[a] | 1.24±0.46[a] | 20.87±13.90 | 21.97±17.70 |
| PDR | 7.79±4.28 | 2.16±1.83[a] | 5.55±1.71[a] | 3.24±1.46[a] | 1.20±037 | 18.05±8.50 | 20.83±13.44 |
| P | 0.5869 | 0.0007 | 0.0000 | 0.0091 | 0.0242 | 0.4778 | 0.1157 |
| Group | FPG | TG | TC | LDL | HDL | [a]ST | [a]LT |
| DNR | 8.17±3.12 | 1.32±0.64 | 4.15±1.07 | 2.44±0.85 | 1.06±032 | 20.36±11.68 | 23.46±17.48 |
| NPDR | 7.99±3.01 | 2.29±1.76[a] | 5.11±1.39[a] | 3.12±1.22[a] | 1.24±0.46[a] | 20.87±13.90 | 21.97±17.70 |
| PDR | 7.79±4.28 | 2.16±1.83[a] | 5.55±1.71[a] | 3.24±1.46[a] | 1.20±037 | 18.05±8.50 | 20.83±13.44 |
| P | 0.5869 | 0.0007 | 0.0000 | 0.0091 | 0.0242 | 0.4778 | 0.1157 |

a: Compared with DNR group $P<0.05$.

b: Compared with NPDR group $P<0.05$.

NPDR or PDR compared to DNR individuals ($P<0.0001$, Table 1). Compared to NPDR subjects, the glucose-6-phosphate dehydrogenase (G6PD) activity had reduced significantly in PDR subjects ($P<0.05$).

## 3.2. G6PD activity was negatively correlated with inflammatory factor secretion in DR patients

Levels of interleukin-6 (IL-6), Levels of interleukin-8 (IL-8) and tumor necrosis factor-α (TNF-α) increased in PDR and NPDR subjects ($P<0.05$, Fig 1). G6PD activity level was negatively correlated with IL-6, IL-8, TNF-α, cholesterol, and low-density lipoprotein (r = -0.1625, -0.1808, -0.1865, -0.1747, r = -0.1807, $P<0.05$, Table 2). Multiple stepwise regression analysis with G6PD activity as the dependent variable and TNF-α, IL-6, IL-8, body mass index (BMI), glomerular filtration rate (GFR), low-density lipoprotein (LDL), and cholesterol as independent variables showed that TNF-α, IL-6, and LDL were independent influencing factors of G6PD activity ($R^2$ = 0.131, F = 7.393, $P<0.01$, Table 2). The severity of retinopathy was set as the dependent variable, including no DR (DNR), non-proliferative DR (NPDR), and proliferative DR (PDR). All related factors were used as independent variables for ordered logistic regression analysis. The results showed that G6PD, triglyceride, cholesterol, IL-8, TNF-α, and macular edema were influencing factors of DR in T2DM patients (Table 3).

## 3.3. The expression of G6PD reduced in the retina of STZ-induced rats

After establishment of diabetic model for 12 weeks, the rat retinas were harvested for western blot and immunofluorescence staining. Western blot showed that the expression of G6PD in the DR group significantly decreased compared to the control group ($P<0.05$, Fig 2A and 2B). The distribution of G6PD in STZ-induced rat retina was determined by immunofluorescence staining, G6PD was found distributing throughout normal rat retinal layer and high co-expressing with endothelium cell marker CD31 (Fig 2C). However, the expression of G6PD in retinal tissue of DR mice was significantly decreased (Fig 2C). Moreover, the co-localization between G6PD and CD31 reduced in DR compared to the control group (Fig 2C–2F).

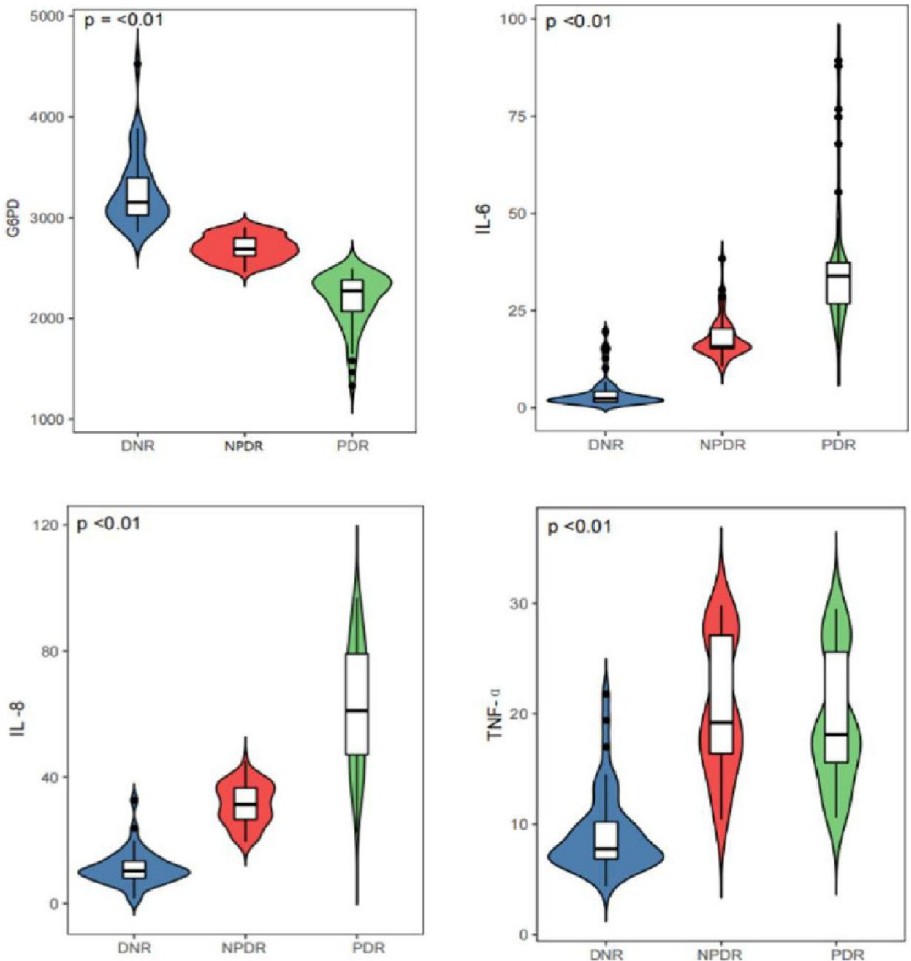

**Fig 1. Comparison of G6PD activity, IL-6, IL-8 and TNF-α among the three groups.**

## 4. Discussion

The typical pathophysiological feature of DR is neurovascular unit dysfunction, mainly including microangiopathy and retinal neurodegeneration [7]. The main manifestations of microangiopathopathy include retinal vascular endothelial cell injury, vascular permeability change, formation of microaneurysms, capillary occlusion, neovascularization, fibrovascular proliferation, etc. [8, 9]. In our in vitro experiments, G6PD and CD31 exhibit fluorescence co-localization expression in the rat retina. This indicates the presence of common expression regions for these two proteins in the retina. Specifically, in the DR group, the expression of G6PD is decreased and partially co-expressed with the CD31 marker. CD31 is a well-known vascular

**Table 2. Stepwise regression analysis of G6PD relating factor.**

| Variable | β | SE | β' | t | P |
|---|---|---|---|---|---|
| Constant | 2394.758 | 223.868 | | 10.697 | 0.000 |
| TNF-α | -19.403 | 7.664 | -0.197 | -2.532 | 0.012 |
| BMI | 21.081 | 8.197 | 0.198 | 2.572 | 0.011 |
| LDL | -58.380 | 23.852 | -0.190 | -2.448 | 0.016 |

**Table 3. Logistic regression analysis of the influencing factors of DR.**

| Variable | β | SE | Waldχ² | P | OR | 95%CI(L) | 95%CI(H) |
|---|---|---|---|---|---|---|---|
| G6PD | -0.002 | 0.001 | 11.203 | 0.001 | 1.159 | -0.003 | -0.001 |
| TG | 0.297 | 0.134 | 4.878 | 0.027 | 1.35 | 0.033 | 0.560 |
| TC | 0.380 | 0.184 | 4.256 | 0.039 | 1.46 | 0.019 | 0.742 |
| IL-$ | 0.82 | 0.34 | 5.721 | 0.017 | 1.09 | 0.015 | 0.149 |
| TNF-a | 0.148 | 0.057 | 6.798 | 0.009 | 1.16 | 0.037 | 0.260 |
| NCSME | -2.254 | 0.550 | 16.795 | 0.000 | 0.10 | -3332 | -1.176 |

marker, suggesting that this co-localization expression implies the involvement of G6PD in aspects such as neovascularization and vascular function regulation in the progression of DR disease.

Two typical features of retinal neurodegeneration are neuronal apoptosis and glial cell proliferation [10]. Currently, many hypotheses exist surrounding the occurrence and development mechanisms of DR, including polyol pathway activation, advanced glycosylation end products, oxidative stress, chronic inflammatory reactions, and apoptosis [11]. G6PD levels vary across different DR stages and normal controls. G6PD not only participates in DR onset and progression but also reflects DR severity [12]. As a key enzyme regulating bodily oxidative-antioxidant balance, G6PD is closely related to multiple pathologies. This study found G6PD activity declines across DNR, NPDR, and PDR patient groups. Western blot results also indicate G6PD expression significantly decreases in retinal endothelial cells of DR patients versus controls [7]. Recently, chronic inflammation was shown to drive pathological changes in retinal neurovascular units, with DR argued to be a persistent, chronic inflammatory disease. Proinflammatory factor release and leukocyte adhesion constitute initial DR onset events, disrupting the blood-retinal barrier to promote microaneurysms and retinal leakage [8].

General elevation of inflammatory factors was observed in DR patients, including IL-1β, IL-6, IL-8, and TNF-α. Early in inflammation development, leukocyte adhesion and aggregation can occur in retinal capillaries [8]. This leads to vascular occlusion and decreased perfusion, resulting in hypoxia. The hypoxic microenvironment can exacerbate mitochondrial dysfunction and increase reactive oxygen components, further promoting inflammation [9, 10, 14, 15]. Thus, high glucose and hypoxia may decrease G6PD activity, and reduced G6PD activity can directly promote inflammatory factor expression and aggravate the inflammatory response, ultimately causing endothelial cell damage and death. Therefore, the decrease in G6PD in diabetic retinopathy may directly damage retinal vascular endothelial cells. If we can find drugs that activate G6PD, these drugs could potentially reduce the levels of angiogenic factors and inflammatory factors under hypoxic stimulation through G6PD mediation, thereby improving endothelial cell damage. In the future, we could consider using high-throughput screening methods to identify compounds that activate G6PD and evaluate their effects in both in vitro and in vivo models. The focus should be on selecting drugs that effectively enhance G6PD activity and offer protective effects on the retina. Additionally, further research is needed to investigate how G6PD activation specifically impacts angiogenic factors and inflammatory factors in the retina. Techniques such as transcriptomics and proteomics could be employed to reveal the specific roles of G6PD in these processes [12, 16, 17]. Overall, treatments targeting G6PD have the potential to provide new approaches and strategies for the intervention of diabetic retinopathy (DR). Future research should comprehensively consider aspects such as drug screening, mechanism analysis, animal experiments, and clinical validation to achieve effective treatment for DR. However, before the research findings are translated into clinical practice, thorough clinical trials are necessary to verify their safety and

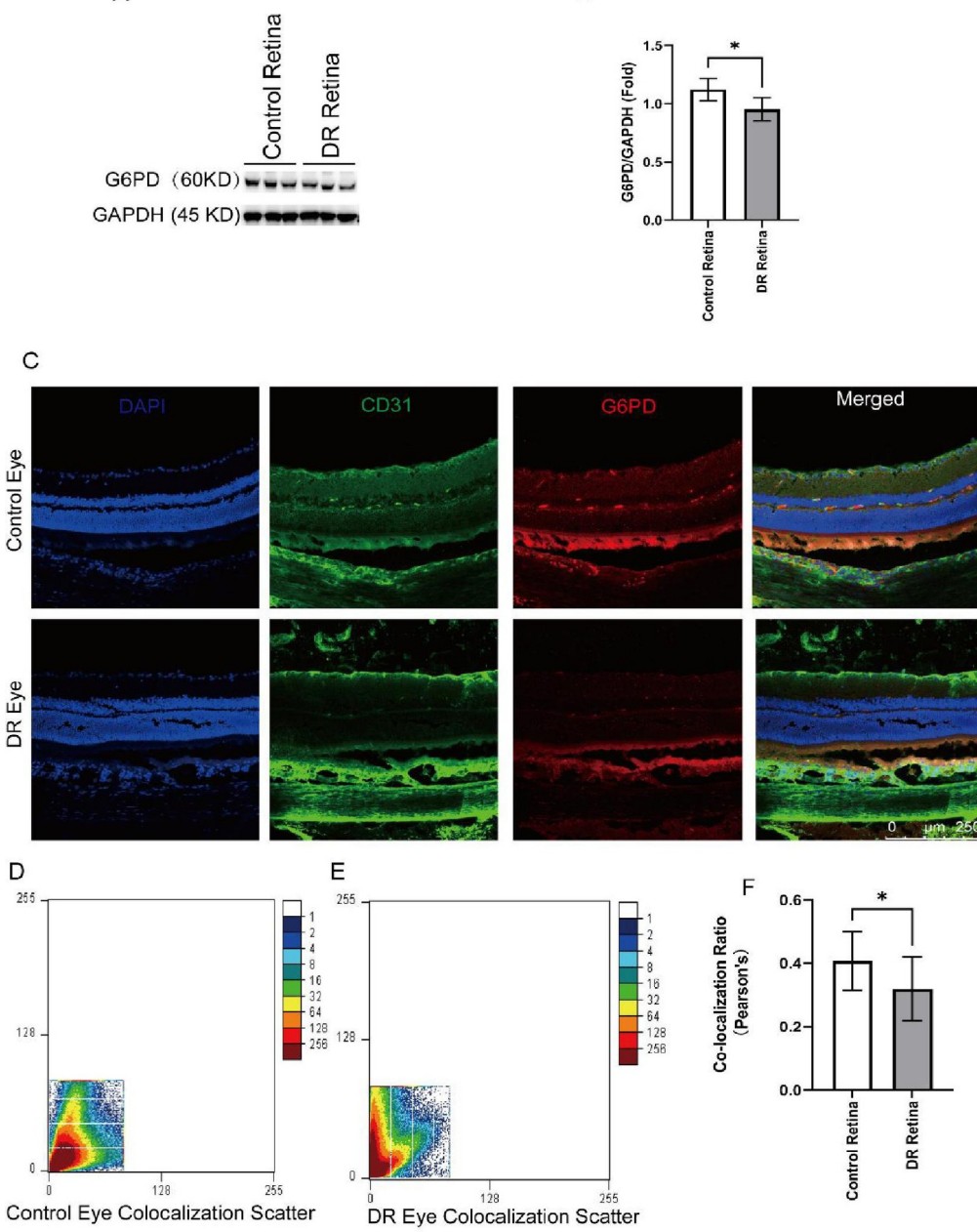

**Fig 2. The expression of G6PD reduced in the retina of STZ-induced rats. (A)** Expression of G6PD in STZ-induced rats were evaluated by western blot. **(B)** The detailed expression level of G6PD normalized to GAPDH. Data were presented as means ± SD. *$P < 0.05$. **(C)** Immunofluorescence images of G6PD and CD31. **(D-E)** Scatter plot of co-localization between G6PD and CD31 in control **(D)** and diabetic retina **(E)**. **(F)** Co-localization between G6PD and CD31 were evaluated by Pearson's correlation coefficient. Data were presented as means ± SD. *$P < 0.05$. (*$P < 0.05$, **$P < 0.01$, ***$P < 0.001$).

effectiveness. The trials should adhere to ethical standards, protect the rights of participants, and provide clear results and follow-up measures to participants at the end of the trial.

Correlation analysis also found G6PD was negatively correlated with IL-6, IL-8, TNF-α, cholesterol, and low-density lipoprotein, suggesting G6PD may have anti-inflammatory effects

in T2DM. Multiple linear regression showed TNF-α and low-density lipoprotein were independent influencing factors of G6PD, suggesting G6PD expression was not only affected by glucose and lipid metabolism, but also possibly by inflammatory factors. Interaction between G6PD and inflammatory cytokines may be one cause of DR in T2DM patients. Although we have strictly controlled potential confounding factors during sample inclusion, some unavoidable factors such as dietary patterns, lifestyles, and drug usage history might still influence G6PD and systemic inflammatory factor activity, thereby affecting our results. However, according to past research findings, these impacts are considered minimal.

Currently, few reports exist on the relationship between G6PD and DR domestically and abroad. Recent studies have found that MNPDR patients have lower G6PD activity levels compared to normal individuals [18–20]. Studies have suggested that direct knockout of G6PD in hepatocytes, liver cancer cells, and fibroblasts leads to a significant increase in IL-8 expression [21]. Stimulation with IL-1β can result in increased glucose uptake by human aortic smooth muscle cells (HASMC) under high glucose conditions, which is then metabolized through the pentose phosphate pathway (PPP), leading to excessive activation of NADPH oxidase, ultimately resulting in increased free radicals and activation of downstream pro-inflammatory signaling pathways [22]. In human adipose tissue, the mRNA level of G6PD is positively correlated with the expression of macrophage markers (CD68 and Mac1) and MCP-1. G6PD in macrophages can promote the activation of MAPK and NF-κB pathways [23]. Based on the above, G6PD is widely involved in regulating cell proliferation. However, its role in regulating oxidative stress and inflammation remains quite complex and controversial. This study also showed that compared with the DNR group, the level of G6PD was decreased in the NPDR and PDR groups. Logistic regression analysis showed that G6PD was an influencing factor of DR, indicating that G6PD may played an important role in the development of DR. Based on the existing research, there are limited studies on the role and mechanisms of G6PD in diabetic retinopathy (DR). However, the review of these studies suggests that G6PD plays an indispensable role in regulating oxidative stress and inflammation. Our research results are consistent with these findings, further confirming the significant role of G6PD in the occurrence and development of DR. Hypoactive or hyperactive G6PD has been linked to type II diabetes pathology [24]. In endothelial cells, G6PD modulates vascular endothelial growth factor and represents a novel angiogenic regulator [25]. As a chronic inflammatory disease, inflammatory factor generation may play a pivotal role in DR pathogenesis. The interaction between G6PD and inflammation factors remains unexplored. Therefore, we hypothesize that G6PD may affect DR progression through modulation of inflammation factor production, such as IL-6, IL-8, TNF-α. As the first enzyme of the pentose phosphate pathway, which provides the principal intracellular source of NADPH, G6PD may affect inflammation factors via complex oxidative stress signaling pathways.

We hypothesize that targeting G6PD and enhancing its activity may have a beneficial effect on improving diabetic retinopathy. Therefore, in T2DM patients, increasing G6PD activity levels may regulate the occurrence and development of DR, so as to play a role in improving DR, which may provide a new direction for the prevention and treatment of DR. But this hypothesis needs further research to confirm. This study has certain limitations: Firstly, cross-sectional studies can only provide information about the status and characteristics of a population at a specific point in time and cannot establish causal relationships between G6PD, inflammatory factors, and DR. Secondly, due to time and resource constraints, the sample size is relatively small, which affects the comprehensiveness of the study. Therefore, future research should aim to a larger and more diverse cohort, use longitudinal methods, and incorporate objective evaluation indicators to improve the study. For instance, conducting long-term follow-ups on patients to observe whether G6PD activity changes with the progression of DR. In

conclusion, G6PD may play an important role in the occurrence and development of T2DM complicated with DR. Its decreased expression in DR is not only influenced by lipid metabolism but also possibly by inflammatory factors. Further study of the mechanism of G6PD in DR may provide important clues and basis for the prevention and treatment of T2DM complicated with DR.

## Supporting information

**S1 File.**
(ZIP)

## Author Contributions

**Conceptualization:** Jun Wang, Ming-Ming Yang.

**Data curation:** Dan Liu, Chuchu Cheng, Lan Zhou, Jun Wang, Ming-Ming Yang.

**Formal analysis:** Dan Liu, Jun Wang, Ming-Ming Yang.

**Funding acquisition:** Ming-Ming Yang.

**Investigation:** Dan Liu, Chuchu Cheng, Tao Zi.

**Methodology:** Dan Liu, Chuchu Cheng, Ming-Ming Yang.

**Project administration:** Lan Zhou, Tao Zi.

**Resources:** Chuchu Cheng, Lan Zhou, Qiqiao Zeng, Hongyan Sun.

**Software:** Qiqiao Zeng, Gongyi Chen.

**Supervision:** Qiqiao Zeng, Gongyi Chen, Hongyan Sun.

**Validation:** Dan Liu, Cunzi Li.

**Visualization:** Tao Zi, Gongyi Chen, Hongyan Sun, Cunzi Li.

**Writing – original draft:** Dan Liu, Chuchu Cheng.

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
