## [Decision Letter · Decision Letter 0]

29 Aug 2024

PONE-D-24-26543Activity of glucose-6-phosphate dehydrogenease and its correlation with inflammatory factors in diabetic retinopathyPLOS ONE

Dear Dr. Liu,

Thank you for submitting your manuscript to PLOS ONE. After careful consideration, we feel that it has merit but does not fully meet PLOS ONE’s publication criteria as it currently stands. Therefore, we invite you to submit a revised version of the manuscript that addresses the points raised during the review process, specifically those of reviewer 2.

We look forward to receiving your revised manuscript.

Kind regards,

Benedikt Ley, PhD

Academic Editor

PLOS ONE

Journal Requirements: When submitting your revision, we need you to address these additional requirements. 1. Please ensure that your manuscript meets PLOS ONE's style requirements, including those for file naming. The PLOS ONE style templates can be found at https://journals.plos.org/plosone/s/file?id=wjVg/PLOSOne_formatting_sample_main_body.pdf and https://journals.plos.org/plosone/s/file?id=ba62/PLOSOne_formatting_sample_title_authors_affiliations.pdf 2. Thank you for stating the following financial disclosure: "This study was supported in part by Shenzhen Science and Technology Project (No. JCYJ20220818102603007), and the General Project of the Shenzhen Natural Science Foundation (No. JCYJ20210324113808023 and JCYJ20220530152813030) Ming-Ming Yang received each award.YMM designed the study and revised the manuscript." Please state what role the funders took in the study.  If the funders had no role, please state: ""The funders had no role in study design, data collection and analysis, decision to publish, or preparation of the manuscript."" If this statement is not correct you must amend it as needed. Please include this amended Role of Funder statement in your cover letter; we will change the online submission form on your behalf. 3. We note that your Data Availability Statement is currently as follows: All relevant data are within the manuscript and its Supporting Information files. Please confirm at this time whether or not your submission contains all raw data required to replicate the results of your study. Authors must share the “minimal data set” for their submission. PLOS defines the minimal data set to consist of the data required to replicate all study findings reported in the article, as well as related metadata and methods (https://journals.plos.org/plosone/s/data-availability#loc-minimal-data-set-definition). For example, authors should submit the following data: - The values behind the means, standard deviations and other measures reported;- The values used to build graphs;- The points extracted from images for analysis. Authors do not need to submit their entire data set if only a portion of the data was used in the reported study. If your submission does not contain these data, please either upload them as Supporting Information files or deposit them to a stable, public repository and provide us with the relevant URLs, DOIs, or accession numbers. For a list of recommended repositories, please see https://journals.plos.org/plosone/s/recommended-repositories. If there are ethical or legal restrictions on sharing a de-identified data set, please explain them in detail (e.g., data contain potentially sensitive information, data are owned by a third-party organization, etc.) and who has imposed them (e.g., an ethics committee). Please also provide contact information for a data access committee, ethics committee, or other institutional body to which data requests may be sent. If data are owned by a third party, please indicate how others may request data access. 4. Your ethics statement should only appear in the Methods section of your manuscript. If your ethics statement is written in any section besides the Methods, please move it to the Methods section and delete it from any other section. Please ensure that your ethics statement is included in your manuscript, as the ethics statement entered into the online submission form will not be published alongside your manuscript. 5. PLOS ONE now requires that authors provide the original uncropped and unadjusted images underlying all blot or gel results reported in a submission’s figures or Supporting Information files. This policy and the journal’s other requirements for blot/gel reporting and figure preparation are described in detail at https://journals.plos.org/plosone/s/figures#loc-blot-and-gel-reporting-requirements and https://journals.plos.org/plosone/s/figures#loc-preparing-figures-from-image-files. When you submit your revised manuscript, please ensure that your figures adhere fully to these guidelines and provide the original underlying images for all blot or gel data reported in your submission. See the following link for instructions on providing the original image data: https://journals.plos.org/plosone/s/figures#loc-original-images-for-blots-and-gels.   In your cover letter, please note whether your blot/gel image data are in Supporting Information or posted at a public data repository, provide the repository URL if relevant, and provide specific details as to which raw blot/gel images, if any, are not available. Email us at plosone@plos.org if you have any questions.

Reviewers' comments:

Reviewer's Responses to Questions

**Comments to the Author**

1. Is the manuscript technically sound, and do the data support the conclusions?

Reviewer #1: Partly

Reviewer #2: Yes

2. Has the statistical analysis been performed appropriately and rigorously? 

Reviewer #1: Yes

Reviewer #2: Yes

3. Have the authors made all data underlying the findings in their manuscript fully available?

Reviewer #1: No

Reviewer #2: Yes

4. Is the manuscript presented in an intelligible fashion and written in standard English?

Reviewer #1: Yes

Reviewer #2: Yes

5. Review Comments to the Author

Reviewer #1: Diabetic retinopathy (DR) is characterised by changes in the microvasculature of the retina secondary to vascular and inflammatory etiology induced by chronic hyperglycemia.

1. Can you explain why you did a clinical assessment of glucose-6-phosphate dehydrogenase (G6PD)

activity in DR and its correlation with inflammatory factors and again an experiment in rats?

2. kindly add the clinical significance of your work.

Reviewer #2: Reviewer’s Comments on Manuscript PONE-D-24-26543

Title: Activity of glucose-6-phosphate dehydrogenase and its correlation with inflammatory factors in diabetic retinopathy

Authors: Dan Liu, Chuchu Cheng, Lan Zhou, et al.

The manuscript titled "Activity of glucose-6-phosphate dehydrogenase and its correlation with inflammatory factors in diabetic retinopathy" addresses a critical area of research, exploring the relationship between G6PD activity and diabetic retinopathy (DR). The study is well-conceived, with robust methodologies and significant findings that contribute valuable insights to the field. However, several areas require minor revisions to enhance the clarity, depth, and overall impact of the paper.

Specific Comments and Suggestions:

1. Longitudinal Data and Study Design:

Comment: The study provides cross-sectional data which is valuable, but the lack of longitudinal follow-up limits the ability to observe the progression of DR and the long-term impact of G6PD activity.

Suggestion: Consider adding a brief discussion on the limitations posed by the cross-sectional design and suggest how future research could address these limitations by incorporating longitudinal studies.

2. Sample Size and Generalizability:

Comment: While the sample size is adequate for the current study, a larger sample could enhance the generalizability of the findings.

Suggestion: Please acknowledge this limitation in the discussion section and propose that future studies should aim to include a larger and more diverse cohort.

3. Confounding Variables:

Comment: The study does not extensively address potential confounding factors, such as diet, medication, or lifestyle, which could influence both G6PD activity and DR progression.

Suggestion: Include a brief discussion of these potential confounders and their possible impact on the results, even if they were controlled for or deemed not significant.

4. Mechanistic Insights:

Comment: The study establishes correlations but does not delve deeply into the mechanistic pathways linking G6PD activity to DR.

Suggestion: Consider expanding the discussion to include possible mechanistic pathways and propose how future studies could explore these mechanisms in more detail.

5. Explanation of Statistical Models:

Comment: The statistical models used are comprehensive, but their complexity might make the analysis less accessible to a broader audience.

Suggestion: Simplify the explanation of these models where possible and provide additional clarification or references for readers who may not be familiar with these statistical techniques.

6. Therapeutic Implications:

Comment: The manuscript hints at the therapeutic potential of modulating G6PD activity but does not explore this in depth.

Suggestion: Expand on the discussion regarding the potential therapeutic implications of your findings and suggest specific avenues for future research that could explore treatments targeting G6PD.

7. Minor Typographical and Formatting Errors:

Comment: There are a few minor typographical errors and inconsistencies in formatting throughout the manuscript.

Suggestion: A thorough proofreading and formatting review should be conducted to correct these minor issues.

8. Figures and Tables:

Comment: The figures and tables are generally well-prepared, but some could benefit from additional labeling or explanations to enhance clarity.

Suggestion: Ensure all figures and tables are fully labeled and consider adding explanatory notes where necessary to aid in the interpretation of the data presented.

9. Ethical Considerations:

Comment: The ethical approvals for both human and animal studies are well-documented, but a more detailed discussion of the ethical implications of your findings would strengthen the manuscript.

Suggestion: Expand on the ethical considerations related to the study’s findings, particularly regarding the potential translation of this research into clinical practice.

10. Discussion of Previous Research:

Comment: The manuscript references relevant literature, but the discussion could benefit from a more critical comparison with previous studies.

Suggestion: Enhance the discussion by critically comparing your findings with previous research and highlighting how your study adds to the existing body of knowledge.

Conclusion:

Overall, this is a well-executed study that makes a valuable contribution to the field of diabetic retinopathy research. The suggested revisions are intended to clarify and strengthen the manuscript, making it even more impactful. I look forward to seeing the revised version of this important work.

6. PLOS authors have the option to publish the peer review history of their article (what does this mean?). If published, this will include your full peer review and any attached files.

Reviewer #1: No

Reviewer #2: No

---

## [Author Response · Author response to Decision Letter 0]

19 Sep 2024

Dear editor and reviewers,

We would like to express our sincere appreciation for your letter and the reviewers’ constructive comments concerning our article entitled “Activity of glucose-6-phosphate dehydrogenease and its correlation with inflammatory factors in diabetic retinopathy” (Manuscript No: PONE-D-24-26543). We have revised our manuscript accordingly. In this revised version, changes to our manuscript are highlighted within the document using red-colored text. Point-by-point responses to the reviewers’ comments are listed below. Once again, we appreciate your kind consideration of our submission. 

With lots of thanks,

Dan Liu

Response to reviewer #1

Comment 1: Can you explain why you did a clinical assessment of glucose-6-phosphate dehydrogenase (G6PD) activity in DR and its correlation with inflammatory factors and again an experiment in rats?

Response: We welcome the reviewer's careful reading and questions. The pathogenesis of diabetic retinopathy (DR) is diverse, and G6PD is a key enzyme in sugar metabolism and oxidative stress mechanisms, playing an important role in DR development. The oxidative stress pathway cannot function without inflammatory factors, participants in the same pathway. Therefore, we aimed to study their relationship to investigate DR pathogenesis. Through clinical trials, we found a certain correlation between the two. Moreover, from the glucose metabolism perspective, the occurrence and development of DR study is insufficient, with most research limited to clinical epidemiological surveys. To explore if the pentose phosphate pathway, a key branch of glucose metabolism, may causally regulate DR occurrence and development, we constructed a STZ-induced diabetic rat model and analyzed the correlation between G6PD and related inflammatory factors throughout the process.

Comment 2: Kindly add the clinical significance of your work.

Response: Thank you for your suggestions. Exploring the causal relationship between G6PD and DR (diabetic retinopathy) could provide new insights and treatment directions. We have added a discussion of the clinical significance of this study in the article's discussion section, highlighted in red font. Please refer to lines 244-250 of the article.

Response to reviewer #2

Comment 1: The study provides cross-sectional data which is valuable, but the lack of longitudinal follow-up limits the ability to observe the progression of DR and the long-term impact of G6PD activity.

Response: We would like to express our deepest gratitude for your insightful feedback, which has played a pivotal role in enhancing the quality of our research. In the discussion section of the article, we have meticulously acknowledged the inherent limitations of cross-sectional studies and have provided a comprehensive set of relevant optimization suggestions to mitigate these drawbacks. You can find these detailed insights in the lines 291-298 of the article.

Comment 2: While the sample size is adequate for the current study, a larger sample could enhance the generalizability of the findings.

Response: We greatly appreciate your comments and suggestions. We have acknowledged the limitation regarding sample size in the article and provided relevant solutions. Please refer to the discussion section of the article for details.

Comment 3: The study does not extensively address potential confounding factors, such as diet, medication, or lifestyle, which could influence both G6PD activity and DR progression.

Response: Thank you for your comments. In experimental research, confounding factors can indeed affect the results, so controlling these factors is crucial. Therefore, we aim to eliminate or control confounding variables during the experimental design phase and use appropriate statistical methods during the data analysis phase to adjust for the impact of these factors. This approach helps improve the reliability and validity of the research findings. We are aware of this limitation and have added a brief discussion of these potential confounding factors and their possible impact on the results in the article, as detailed in lines 256-260.

Comment 4: The study establishes correlations but does not delve deeply into the mechanistic pathways linking G6PD activity to DR.

Response: Thank you for your valuable suggestion. Hypoactive or hyperactive G6PD has been linked to the pathology of type II diabetes, but there are no clear reports on the mechanisms by which G6PD influences the occurrence and progression of DR (diabetic retinopathy) currently. In endothelial cells, it is now clear that G6PD would modulate vascular endothelial growth factor and represent a novel angiogenic regulator. As a chronic inflammatory disease, inflammatory factor generation might also play a pivotal role in the pathogenesis of DR. The interaction between G6PD and inflammation factors remains an area for further investigation. Therefore, we hypothesize that G6PD may affect the progression of DR through modulating the production of inflammation factors such as IL-6, IL-8, TNF-α. As the first enzyme of the pentose phosphate pathway, which is the principal intracellular source of NADPH, G6PD may affect these inflammation factors through a series of complex oxidative stress signal pathway, which will be a huge challenge to elucidate clearly. Therefore, we have added possible mechanisms and future research plans in the discussion section, as detailed in lines 263-286 of the article.

Comment 5: The statistical models used are comprehensive, but their complexity might make the analysis less accessible to a broader audience.

Response: We greatly appreciate your comments. We have added explanations related to statistical methods in the methodology section to help readers better understand the article.

Comment 6: The manuscript hints at the therapeutic potential of modulating G6PD activity but does not explore this in depth.

Response: Thank you for your careful reading and suggestions. Following your suggestion we have included the potential therapeutic significance of G6PD and specific future research directions in the article. Please refer to lines 234-250 for the details.

Comment 7: There are a few minor typographical errors and inconsistencies in formatting throughout the manuscript.

Response: Thank you for your careful review. We apologize for our oversight. Based on your feedback, we have thoroughly proofread and formatted the article.

Comment 8: The figures and tables are generally well-prepared, but some could benefit from additional labeling or explanations to enhance clarity.

Response: Thank you for your comments. We have rechecked all the figures in the article to ensure that each one has proper labels and necessary annotations.

Comment 9: The ethical approvals for both human and animal studies are well-documented, but a more detailed discussion of the ethical implications of your findings would strengthen the manuscript.

Response: Thank you for your feedback. We have incorporated ethical considerations regarding the potential translation of this research into clinical practice, as suggested. Please refer to lines 244-250 of the article for details.

Comment 10: The manuscript references relevant literature, but the discussion could benefit from a more critical comparison with previous studies.

Response: Thank you for your suggestion. We have added a discussion of the current controversies and consensus in related research to the discussion section of the article, and we have also detailed how our study contributes to and impacts the current research field.

---

## [Decision Letter · Decision Letter 1]

8 Oct 2024

Activity of glucose-6-phosphate dehydrogenease and its correlation with inflammatory factors in diabetic retinopathy

PONE-D-24-26543R1

Dear Dr. Liu,

We’re pleased to inform you that your manuscript has been judged scientifically suitable for publication and will be formally accepted for publication once it meets all outstanding technical requirements.

Kind regards,

Benedikt Ley, PhD

Academic Editor

PLOS ONE

Additional Editor Comments (optional):

Reviewers' comments:

Reviewer's Responses to Questions

**Comments to the Author**

1. If the authors have adequately addressed your comments raised in a previous round of review and you feel that this manuscript is now acceptable for publication, you may indicate that here to bypass the “Comments to the Author” section, enter your conflict of interest statement in the “Confidential to Editor” section, and submit your "Accept" recommendation.

Reviewer #1: All comments have been addressed

Reviewer #2: All comments have been addressed

2. Is the manuscript technically sound, and do the data support the conclusions?

Reviewer #1: Yes

Reviewer #2: Yes

3. Has the statistical analysis been performed appropriately and rigorously? 

Reviewer #1: Yes

Reviewer #2: Yes

4. Have the authors made all data underlying the findings in their manuscript fully available?

Reviewer #1: Yes

Reviewer #2: Yes

5. Is the manuscript presented in an intelligible fashion and written in standard English?

Reviewer #1: Yes

Reviewer #2: Yes

6. Review Comments to the Author

Reviewer #1: The review comments have been addressed by the authors. The revised manuscript reads well. Further studies may show if targeting G6PD may have a beneficial effect on

diabetic retinopathy

Reviewer #2: Following a comprehensive review, the authors have revised the entire manuscript, effectively addressing all provided comments.

I have carefully considered their responses and the modifications made.

The authors have successfully integrated the feedback, enhancing the clarity and quality of the manuscript. The updated information aligns well with the expectations outlined in the previous review. Each revision demonstrates a clear understanding of the feedback provided. The authors' efforts have resulted in significant improvements to the overall content.

Therefore, I am pleased to recommend the acceptance of this manuscript for publication.

7. PLOS authors have the option to publish the peer review history of their article (what does this mean?). If published, this will include your full peer review and any attached files.

Reviewer #1: No

Reviewer #2: No

---

## [Editor Report · Acceptance letter]

14 Oct 2024

PONE-D-24-26543R1 

PLOS ONE

Dear Dr. Liu, 

I'm pleased to inform you that your manuscript has been deemed suitable for publication in PLOS ONE. Congratulations! Your manuscript is now being handed over to our production team.

Kind regards, 

on behalf of

Dr Benedikt Ley 

Academic Editor

PLOS ONE